# MODEL-AGNOSTIC TEXT CONDENSATION WITH COHERENCE AWARENESS

## ABSTRACT

Data condensation has emerged as a promising technique for improving training efficiency. However, it remains challenging to produce a small synthetic text set that retains its utility for use with language models. Existing approaches are typically model-specific and often focus only on generating readable text, which limits their applicability to text understanding tasks (e.g., classification). In this work, we propose a model-agnostic text condensation framework with coherence awareness. Our method synthesizes a compact set of representative texts by modeling in the semantic embedding space while enforcing coherence constraints when converting them back into the input space. This model-agnostic design allows the condensed data to be used for training or adapting a wide range of models without retraining the condensation pipeline. Experiments on diverse language understanding and reasoning benchmarks show that our method outperforms state-of-the-art text condensation techniques. Our work highlights the importance of preserving textual coherence in dataset condensation and opens new avenues for efficient and reusable data preparation across models.

## 1 INTRODUCTION

The rapid advancements in language models have been significantly driven by the availability of large-scale text datasets. Although larger datasets often yield better performance, there is increasing recognition that smaller but higher-quality data can be more effective (Gunasekar et al., 2023). This motivates the study of data condensation (or distillation), which has been extensively explored in the image domain but remains only a few for text. Recent efforts Li & Li (2021); Xie et al. (2024); Tao et al. (2024); Nguyen et al. (2025); Maekawa et al. (2025a) have attempted to adapt image-based condensation techniques to textual data, addressing challenges such as discreteness of input, variable sequence lengths, and readability. Since textual data can be used for training, fine-tuning, and in-context learning across diverse (large) language models, we propose to study the Model-agnostic Text Condensation (MaTC) problem.

MaTC essentially requires generating *in-distribution* condensed samples, since it is agnostic to downstream models and the textual information aggregated from training samples cannot be propagated through gradients (Maekawa et al., 2025b). Given a certain number of generated samples, it must satisfy the following fundamental properties:

(1) Representativeness. Condensed text should reflect the global distribution of the original dataset.

(2) Diversity. Condensed text should ensure coverage of different modes and prevents redundancy.

(3) Coherence. Each condensed sample remains logically consistent and semantically complete.

Representativeness and diversity have been recognized in existing data condensation works. Gu et al. (2024) defined representativeness as the cosine similarity between original and condensed samples in the embedding space, and diversity as maximizing the pairwise distances among synthetic samples. In contrast, Chan-Santiago et al. (2025) advocated improving diversity by clustering within each image class and using the cluster centers as anchors to regularize the denoising process in diffusion models. While these definitions and insights were proposed for images, we extend them to the text domain. To improve the downstream usability of condensed text, we introduce coherence, shown as Fig. 1, which goes beyond simple readability Tao et al. (2024). While readability ensures that a

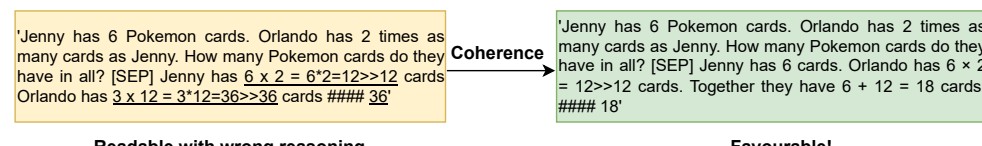

**Readable with wrong reasoning**          **Favourable!**

Figure 1: Example of condensed text sample on GSM8K. The left box shows the inverted readable sample with incorrect reasoning underlined, and the right box shows the coherence-refined version.

text is grammatically correct and easy to follow, coherence additionally requires logical consistency, structural integrity, and the preservation of semantic relations. This stricter property is particularly crucial for reasoning tasks, where solving a problem depends not only on fluent text but also on the correctness of intermediate steps, the ordering of information, and the use of special tokens (e.g., [SEP]).

We respond the three key properties of text condensation by proposing a new framework. Representativeness and diversity are achieved by optimizing informative particles in a semantic embedding space, ensuring that the condensed set preserves the global distribution of the original data and spreads across different high-density regions. And coherence is enforced in the invert-and-refinement stage, where derived particles are inverted into discrete text and refined with API assistance to ensure logically consistent and structurally sound samples. We name this entire framework as PInR and validate its efficacy on both understanding and reasoning tasks.

Our main contributions can be summarized as follows:

• We are the first to propose text coherence as a key property for model-agnostic text condensation, extending beyond the conventional requirement of human readability, which is particularly critical for reasoning tasks. Together with representativeness and diversity—two properties emphasized in recent work on image condensation—we identify these three as essential and unify them with a distribution approximation angle.

• We propose a new framework that optimizes condensed data by first searching for informative particles in the embedding space, analytically encouraging representativeness and diversity. These particles are then inverted into discrete text, followed by an API-assisted refinement optimization that generate coherent text samples for downstream use.

• We evaluate our method on both understanding and reasoning tasks, where it consistently outperforms state-of-the-art baselines. We further discuss the potential extensions of our framework to privacy-sensitive data and highlight current limitations, laying the groundwork for future research in this direction.

## 2 RELATED WORK

Our review centers on advances in text condensation, with occasional references to image-based works closely related to our method.

### 2.1 CORESET SELECTION

Coreset selection aims to identify a subset of data that achieves performance comparable to the full dataset, and is also referred to as data pruning (Mirzasoleiman et al., 2020). In the text domain, sample selection occurs either during language model pre-training (Wenzek et al., 2020; Azeemi et al., 2023) or during the fine-tuning phase (Nguyen & He, 2025). Most pre-training stage approaches rely on heuristic strategies (Marion et al., 2023), which are not strictly sample-wise but instead operate through sentence-level filtering (Xue et al., 2021). In contrast, research on text condensation for fine-tuning transformer-based language models often leverages downstream models to estimate sample importance, either by measuring downstream performance (Attendu & Corbeil, 2023) or by exploiting strong LLMs as evaluators (Chen et al., 2023). Additional criteria have also been introduced, such as fairness considerations (Zayed et al., 2023) and systematic modeling of inter-sample relationships (Maharana et al., 2023).

## 2.2 DATASET CONDENSATION

The key idea of most previous work on dataset condensation is to train models on synthetic data that can mimic the behavior of training on real data. Sucholutsky & Schonlau (2021) presented an early example of this approach by distilling soft labels. Li & Li (2021) generated human-unreadable numerical data, where the variables are treated as parameters, enabling gradient-descent-based optimization. Maekawa et al. (2025a) further proposed distilling attention labels for fine-tuning transformers, and subsequently train a language model to generate informative samples (Maekawa et al., 2025b). Beyond these methods, which are not agnostic to downstream tasks, recent work on data synthesis (Tao et al., 2024; Cai et al., 2025) can also be viewed within this direction, often with an additional emphasis on privacy concerns (Xie et al., 2024; Yue et al., 2022).

## 3 PRELIMINARIES

**Problem statement.** Consider a large-scale dataset with the training set $\mathcal{T}_o = \{x_i\}^N$, where each sample is a textual sequence[1], collectively prepared for downstream use, e.g., fine-tuning. The problem of model-agnostic text condensation is to synthesize a dataset $\mathcal{T}_s = \{\tilde{x}_j\}^M$ with $M \ll N$ such that $\mathcal{T}_s$ preserves the essential information of $\mathcal{T}_o$ without relying on downstream models. Formally, for any downstream model $\theta$, we would expect $eval(\theta(\mathcal{T}_o)) \sim eval(\theta(\mathcal{T}_s))$, where $\theta(\mathcal{T}_o)$ and $\theta(\mathcal{T}_s)$ are models trained on or conditional upon $\mathcal{T}_o$ and $\mathcal{T}_s$ respectively, and $eval(\cdot)$ denotes the evaluation criterion of interest.

**Distribution approximation.** Suppose each $x_i \in \mathcal{T}_o$ is drawn i.i.d. from a distribution $p$. The synthetic dataset $\mathcal{T}_s$ can be represented as an empirical measure $\hat{q} = \frac{1}{M} \sum_{j=1}^{M} \delta_{\tilde{x}_j}$ where $\delta_{\tilde{x}_j}$ denotes the Dirac measure centered at $\tilde{x}_j$. The condensation objective is then to minimize a distributional distance $d(\hat{q}, p)$, where $d(\cdot, \cdot)$ denotes a distance metric. The objective comes to a Wasserstein approximation studied in image synthesis applications Lin et al. (2024) when $d(\cdot, \cdot)$ is chosen as the Wasserstein distance.

## 4 METHODOLOGY

In response to the requirement that condensed samples should possess three fundamental properties, representativeness, diversity, and coherence, as discussed in Section 1, we propose a two-stage method to address this task.

## 4.1 PARTICLES OPTIMIZATION WITH LANGUAGE MODEL EMBEDDING

As discussed in Section 3, the objective is to approximate the original text distribution $p$ using a simpler surrogate distribution $q$. This problem can be formulated within the framework of variational inference, where the optimal approximation $q^*$ is obtained by minimizing the Kullback–Leibler (KL) divergence from $q$ to $p$, that is $q^* = \arg\min_q \{\text{KL}(q||p) \equiv \mathbb{E}_q[\log q] - \mathbb{E}_q[\log \bar{p}]\}$, with $\bar{p}$ denoting the unnormalized version of $p$. The normalization constant of $p$ is omitted since it is independent of $q$. Based on the Stein's theory of Liu & Wang (2016), we consider an infinitesimal map $T_\xi(\tilde{x}) = \tilde{x} + \xi\phi(\tilde{x})$ which gradually pushes a randomly initial distribution $q_0$ to $q$ with the steepest direction $\phi(\tilde{x})$ through minimizing the KL functional. The the optimal direction can be written in closed form,

$$\phi^*(\cdot) \propto \mathbb{E}_{\tilde{x} \sim q}[k(\tilde{x}, \cdot)\nabla_{\tilde{x}} \log p(\tilde{x}) + \nabla_{\tilde{x}} k(\tilde{x}, \cdot)], \tag{1}$$

where $k(\cdot, \cdot)$ is the scalar kernel in reproducing kernel Hilbert space. This approach however remains intractable due to the difficulty of drawing samples in the discrete text domain. To ensure the condensation process sufficiently informative, we instead consider their representations in a semantic space through a language model embedding, i.e., $e = \psi(x)$, with $\tilde{e}$ representing the embeddings of $\tilde{x}$ accordingly. Now we randomly draw a set of particles $\{\tilde{e}_j\}_{j=1}^M$ and iteratively update each of

---

[1]We slightly abuse the notation $x_i$ as features for text classification tasks, which allows us to condense class-wise samples similarly to how image samples are handled per class; for generation tasks such as Q&A, $x_i$ can instead denote concatenated sequences.

them until convergence, which we refer to as Stein-based particles. Concretely, at $t + 1$-th iteration, each particle in the embedding space can be updated by:

$$\tilde{e}_j^{t+1} \leftarrow \tilde{e}_j^t + \frac{\xi}{M} \sum_{h=1}^{M} [k(\tilde{e}_h^t, \tilde{e}_j^t) \nabla_{\tilde{e}_h^t} \log p(\tilde{e}_h^t) + \nabla_{\tilde{e}_h^t} k(\tilde{e}_h^t, \tilde{e}_j^t)], \tag{2}$$

where $p(\tilde{e})$ represents the target density evaluated at $\tilde{e}$, indicating how the original samples participate the condensation in the embedding space.

We highlight that the two terms inside the summation in Eq. (2) naturally correspond to *representativeness* and *diversity*, respectively. The first term encourages particles to move toward high-density regions of the target distribution $p(e)$ weighted by kernel similarity, thereby guiding them to cover the potential modes of original samples. The second term acts as a repulsive force which push the $M$ particles away from each other. For example, the gradient instanced with RBF kernel is $\nabla_{\tilde{e}_h} k(\tilde{e}_h, \tilde{e}_j) \propto k(\tilde{e}_h, \tilde{e}_j)(\tilde{e}_j - \tilde{e}_h)$, which pushes $\tilde{e}_j$ away from $\tilde{e}_h$ when they are close.

**Implementation.** The target density through the embedding model $\psi$ can be formally expressed as $p(e) = \int_{\mathcal{X}} p(x)\delta(e - \psi(x))dx$. In practice, we can approximate it empirically using the embeddings $\psi(x_i)$ of all training samples $x_i \in \mathcal{T}_o$. The non-parametric method such as kernel density estimation is simple but numerically unstable for high-dimensional embeddings. Gaussian mixture models provide an analytic score function $\nabla_e \log p(e)$, which can be also alternatively trained by score-based models Hyvärinen & Dayan (2005); Sohl-Dickstein et al. (2015). The scalar kernel is chosen by a RBF with the derived gradient form easy to compute. The particles $\{\tilde{e}_j\}_{j=1}^{M}$ can be initialized with randomly sampled embeddings of the original samples when privacy is not concerned. Regarding text condensation for classification tasks, Eq. (2) can be applied in a class-wise manner, seeking sub-modes within each class, similar to the mode-guided data distillation Chan-Santiago et al. (2025). For generation tasks with structure text within per sample, we concatenate all texts into a single sequence separated by [SEP] tokens before obtaining their embeddings. Further details are left to in Appendix A.1.

## 4.2 INVERT-AND-REFINE (INR)

Although operating in the embedding space enables the particles to converge towards informative regions, the optimized embeddings $\tilde{e}$ cannot be transferred across different language models until they are converted into their corresponding texts $\tilde{x}$. Moreover, to enhance the validity of $\tilde{x}$, we introduce $\mathcal{C}$ as a constraint that guarantees its coherence. Given that embedding models tend to produce similar representations for semantically related inputs, we have the following lexicographic optimization problem,

$$\tilde{x}_j = \arg\min_x d(\psi(x), \tilde{e}_j) \;\; s.t. \; x \in \mathcal{C}, \qquad \forall j \in \{1, \ldots, M\} \tag{3}$$

where coherence serves as a must-satisfy condition. Note that cohenrence can be replaced with a weaker condition such as readability Nguyen et al. (2025) if the downstream tasks are not highly sensitive to it (e.g., sentiment analysis). In contrast, for most structure texts tasks, breaking coherence would severely harm a model's reasoning capability when the condensed data are used for training or conditioning. From the view of optimization, searching for a variable-length sequence $\tilde{x}$ from a large vocabulary to "match" a given $\tilde{e}$ remains challenging, especially in the absence of a task-specific coherence critic.

We find out that the above problem can be alternatively decomposed into learning two modules: a decoder that inverts embeddings (particles) into text, and a refiner that enhances the coherence of the generated text. This Invert-and-Refine (InR) can be expressed in a probabilistic form:

$$p(\tilde{x}|\tilde{e}) = \sum_{\tilde{x}_0} p(\tilde{x}_0|\tilde{e})p(\tilde{x}|\tilde{x}_0, \tilde{e}). \tag{4}$$

The decoder denoted by $\omega(\cdot)$ is trained on $\mathcal{T}_o$ using an encoder-decoder transformer architecture with the embedding model $\psi(\cdot)$ serving as the frozen encoder. We follow the implementation of vec2text Morris et al. (2023) for $\omega(\cdot)$, which is instantiated as a recursive conditional generation model (See more details in Appendix A.1). With this approach, the resulting $\tilde{x}_0$ may lack semantic meaningfulness as the updated $\tilde{e}$ through Eq. (2) is new to $\omega(\cdot)$. Fig. x shows a example. The refiner

module adopts a strategic approach that explores the possible variations through a callable API, e.g., GPT-3.5. Specifically, we generate $L$ variations within a small neighborhood of $\tilde{x}_0$ by using a prompt (e.g., "rephrase the given text to be logical with minimal changes"). These variations, denoted as $\tilde{x}'$ are then considered coherent. Among them, we select the sample whose embedding is closest to $\tilde{e}$. By defining $d(\cdot, \cdot)$ as the negative cosine similarity, the output $\tilde{x}$ can be written as $\tilde{x} = \arg\max_{l \in \{1,...,L\}} \cos(\tilde{e}, \psi(\tilde{x}^l))$. In practice, we can perform a multi-step refinement process, then Eq. (4) generalizes to $p(\tilde{x}_T|\tilde{e}) = \sum_{\tilde{x}_0} \sum_{\tilde{x}_1} ... \sum_{\tilde{x}_{T-1}} p(\tilde{x}_0|\tilde{e}) \prod_{t=0}^{T-1} p(\tilde{x}_{t+1}|\tilde{x}_t, \tilde{e})$. In this formulation, since we marginalize over intermediate generation $\tilde{x}_t$ at each step, we may retain the top-$K$ closest variations as seeds for producing the next set of candidate variations.

We refer to the full method as PInR, and Fig. 2 illustrates its overall structure. Given an embedding model $\psi(\cdot)$, PInR trains a score function to guide particle optimization in the embedding space and a decoder $\omega(\cdot)$ that inverts the embeddings to text. The optimized particles are then fed to the trained decoder which produce the initial text sequences. Each optimized embedding $\tilde{e}$ serves as a constraint to ensure that API-assisted refinement remains informative and does not deviate from the 'anchors' that best approximate the original data distribution. A more detailed algorithm is provided in Appendix A.2.

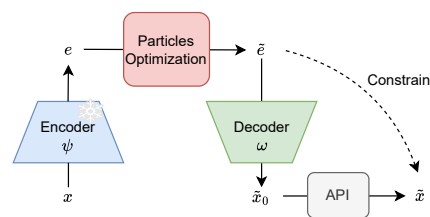

Figure 2: Overview of the PInR framework, where the encoder is the only fixed module.

## 5 EXPERIMENTS

### 5.1 EXPERIMENT SETUP

**Datasets.** We evaluate our PInR on four benchmark datasets: AG-News (Gulli & Sekine, 2005), SST-2 (Wang et al., 2019), GSM8K (Cobbe et al., 2021), and Quora-QuAD (Toughdata, 2023). AG-News and SST-2 are adopted for text understanding tasks, applied in a class-conditional generation manner. Two reasoning-related datasets are employed to validate the necessity of incorporating text coherence into the condensation process: GSM8K for mathematical calculation, and Quora-QuAD for reading comprehension.

**Baselines.** We consider three state-of-the-art methods for model agnostic text condensation. (1) DaLLME (Tao et al., 2024): clustering in the embedding space and inverting cluster centers back to the input space. The number of clusters is set equal to the number of condensed samples. (2) MGD³ (Chan-Santiago et al., 2025): clustering to identify modes in the embedding space, which serve as a regularizer (often within each class) to enhance diversity. This method is adapted from image distillation. (3) Aug-PE (Xie et al., 2024): synthesizing condensed samples that approximate the target distribution by leveraging API outputs. Infinite privacy budget is applied for a fair comparison in settings without privacy constraints. Moreover, we consider selecting a subset of the original samples uniformly at random, with their number equal to that of the condensed set. We denote this method as Random, which serves as a reference and has been validated as a strong baseline in coreset selection (Nguyen & He, 2025).

**Models.** Given an embedding model, the decoders in our method are trained following the procedure of Morris et al. (2023). When optimizing particles, we use nonparametric models to optimize score function, which already yields good performance. For API-based refinement, to avoid concerns that API capabilities may give our method an advantage, we use the same API version as the baseline methods whenever applicable, ensuring a fair comparison.

**Metrics.** For understanding tasks, we fine-tune widely used downstream models including TextRNN (Hu et al., 2020), DistilBERT (Sanh et al., 2019), and T5-base (Raffel et al., 2020) with condensed text samples, and report classification accuracy as the evaluation metric. Regarding reasoning-related tasks, we use Llama-3.1-8B-Instruct (Grattafiori et al., 2024), Phi-3.5-Mini-Instruct (Abdin et al., 2024), and Gemma-2-9B-IT (Team et al., 2024) as downstream models, fine-tuned on the condensed data conditional upon them, or instructed with them as in-context. In addi-

Table 1: Evaluation on the AG-News Dataset (%)

| Downstream Model | Full | Random | DaLLME | MGD$^3$ | Aug-PE | PInR |
|---|---|---|---|---|---|---|
| TextRNN | 92.10 | 74.10 | 67.91 | 72.72 | 68.30 | **78.96** |
| DistilBERT | 94.50 | 78.60 | 86.22 | 84.83 | 80.63 | **87.04** |
| T5-Base | 95.40 | 76.30 | 86.86 | 84.64 | 80.51 | **87.32** |

Table 2: Evaluation on the SST-2 Dataset (%)

| Downstream Model | Full | Random | DaLLME | MGD$^3$ | Aug-PE | PInR |
|---|---|---|---|---|---|---|
| TextRNN | 83.72 | 60.32 | 61.24 | 60.09 | **66.06** | 63.19 |
| DistilBERT | 91.06 | 74.43 | 73.62 | 75.80 | 78.33 | **79.70** |
| T5-Base | 94.15 | 76.61 | 70.30 | 80.16 | 83.26 | **85.21** |

tion, we quantify the similarity between the original and condensed data following the measurements used in Xie et al. (2024).

Throughout all tasks, the best performance is marked in bold, while the second-best is underlined. Except for Random which we report its average results following the convention of recent work (Nguyen & He, 2025), there is no evaluation variance in understanding tasks. In contrast, for reasoning-related datasets we report average results with standard deviations, with performance values multiplied by 100 for clearer presentation. Additional experimental details are provided in Appendix B.2.

## 5.2 Main Results

### 5.2.1 Evaluation with Downstream Tasks

Table 3: Evaluation on the GSM8K Dataset

| Downstream Model | Zero-shot | Type | Random | DaLLME | MGD$^3$ | Aug-PE | PInR |
|---|---|---|---|---|---|---|---|
| Llama-3.1-8B-Instruct | 73.92$_{\pm1.21}$ | FT | 76.95$_{\pm1.16}$ | 75.74$_{\pm1.18}$ | 74.07$_{\pm1.21}$ | 75.13$_{\pm1.19}$ | **77.26$_{\pm1.15}$** |
| | | ICL | **77.55$_{\pm1.15}$** | 74.45$_{\pm1.20}$ | 74.22$_{\pm1.20}$ | 70.35$_{\pm1.26}$ | 75.58$_{\pm1.18}$ |
| Phi-3.5-Mini-Instruct | 59.97$_{\pm1.35}$ | FT | 60.88$_{\pm1.34}$ | 60.42$_{\pm1.35}$ | 60.42$_{\pm1.35}$ | 60.80$_{\pm1.34}$ | **61.56$_{\pm1.34}$** |
| | | ICL | **79.91$_{\pm1.10}$** | 72.25$_{\pm1.23}$ | 72.71$_{\pm1.23}$ | 66.56$_{\pm1.30}$ | 78.24$_{\pm1.13}$ |
| Gemma-2-9B-IT | 73.77$_{\pm1.21}$ | FT | 74.00$_{\pm1.21}$ | **74.07$_{\pm1.21}$** | 73.84$_{\pm1.21}$ | 74.00$_{\pm1.21}$ | **74.07$_{\pm1.21}$** |
| | | ICL | **82.78$_{\pm1.04}$** | 76.72$_{\pm1.16}$ | 76.57$_{\pm1.17}$ | 69.82$_{\pm1.26}$ | 79.61$_{\pm1.11}$ |

We generate 120 and 80 samples for the AG-News and SST-2 datasets, respectively, which correspond to approximately 0.1% of the full training sets, and evaluate accuracy on the original test sets. The details of downstream training configuration are provided in Appendix B.1 to facilitate reproduction of our reported results, and Tables 1 and 2 summarize the corresponding results. On both AG-News and SST-2, we can see that PInR consistently outperforms existing condensation methods across most downstream models. For AG-News, PInR achieves the best accuracy on all three backbones, surpassing Random and clustering-based baselines (DaLLME, MGD$^3$) by a clear margin. Similarly, on SST-2, PInR yields the strongest performance on transformer-based models, and performing slightly worse than Aug-PE on TextRNN. Although the best performance of condensation methods falls short of full-data training, the results confirm that PInR retains much of the original dataset's utility while substantially reducing data size.

On reasoning tasks, to support both fine-tuning (FT) and in-context learning (ICL), we generate 500 samples both on the GSM8K and Quora-QuAD dataset. Regarding ICL, we evaluate under a 3-shot configuration. The evaluation metrics for GSM8K is Exact Match and for Quora-QuAD is Rouge1 (more experimental results in terms of different evaluation metrics are reported in Appendix B). The shaded results in Tables 3 and 4 correspond to tuning Gemma-2-9B-IT with only a small number

Table 4: Evaluation on Quora-QuAD Dataset

| Downstream Model | Zero-shot | Type | Random | DaLLME | MGD[3] | Aug-PE | PInR |
|---|---|---|---|---|---|---|---|
| Llama-3.1-8B-Instruct | $15.44_{\pm0.01}$ | FT | $15.40_{\pm0.00}$ | $\underline{15.68_{\pm0.01}}$ | $15.32_{\pm0.01}$ | $15.40_{\pm0.00}$ | $\mathbf{15.73_{\pm0.01}}$ |
| | | ICL | $\underline{15.64_{\pm0.19}}$ | $13.79_{\pm0.03}$ | $13.63_{\pm0.17}$ | $15.40_{\pm0.15}$ | $\mathbf{17.15_{\pm0.09}}$ |
| Phi-3.5-Mini-Instruct | $11.97_{\pm0.01}$ | FT | $12.01_{\pm0.01}$ | $11.96_{\pm0.00}$ | $\underline{12.05_{\pm0.01}}$ | $11.99_{\pm0.01}$ | $\mathbf{12.09_{\pm0.06}}$ |
| | | ICL | $12.25_{\pm0.25}$ | $\underline{13.18_{\pm1.22}}$ | $13.13_{\pm1.09}$ | $13.11_{\pm1.13}$ | $\mathbf{13.28_{\pm1.13}}$ |
| Gemma-2-9B-IT | $5.73_{\pm0.01}$ | FT | $\mathbf{5.82_{\pm0.01}}$ | $5.71_{\pm0.00}$ | $5.71_{\pm0.01}$ | $5.63_{\pm0.01}$ | $\underline{5.75_{\pm0.01}}$ |
| | | ICL | $11.04_{\pm0.11}$ | $11.40_{\pm0.00}$ | $11.48_{\pm0.01}$ | $\underline{11.51_{\pm0.08}}$ | $\mathbf{11.64_{\pm0.04}}$ |

of samples, a challenging setting where improvements for all methods are limited. On GSM8K (Table 3), PInR consistently achieves competitive or superior performance compared with existing condensation methods across multiple downstream models and training paradigms. For Llama-3.1-8B-Instruct, PInR attains 77.26% (FT) and 75.58% (ICL), both ranking among the best results and slightly improving upon strong baselines such as Random. Note that Random dominates the ICL performance on GSM8K with our method yields the second place. This is because Random is more faithful to original data regarding true mathematical problems. However, our method obtains the best performance on Quora-QuAD dataset in most cases, owing to its inherent linguistic characteristics. The Quora-QuAD dataset spans diverse topics and domains, where a few random samples are insufficient to provide meaningful guidance.

### 5.2.2 EVALUATION WITH SIMILARITY QUANTIFICATION

We employ eight similarity metrics including Fréchet Inception Distance (FID) (Heusel et al., 2017), KL, TV and Wassersteain divergences (Chung et al., 1989), MAUVE score (Pillutla et al., 2021), and Precision, Recall, F1 score (Kynkäänniemi et al., 2019) to evaluate the quality of condensed text across four datasets, and the results are summarized in Table 5. Random often yields strong results, as it can be regarded as an unbiased estimator of the data distribution. Our method consistently ranks among the top approaches, and even in the few cases where it does not achieve a top-two position, its performance remains competitive, with scores closely matching the second-best method. Compared with our approach, the relatively weaker performance of Aug-PE can be attributed to its reliance on distribution matching based on distance metrics. While effective in certain settings, this strategy is highly sensitive to initialization and strongly depends on the diversity of variants contributed by prompt engineering. In contrast, our method directly optimizes within the neighborhood of the inverted text, thereby maintaining robustness without requiring extensive manual design or reliance on diverse prompt variants. This design choice allows our approach to achieve stable performance across datasets with different linguistic and structural characteristics.

### 5.3 UNDERSTANDING THE PERFORMANCE OF PINR

*RQ1: Stein-based particles versus clustering centroids.* When the original data in the embedding space exhibits a clear cluster structure, clustering methods can often achieve satisfactory results, as they theoretically approximate the data distribution under certain assumptions (Canas & Rosasco, 2012). Fig. 3a shows visualizations of particles derived from GSM8K using both Stein-based particles and clustering centroids, where both sets of particles are spread across the data space. However, when only a small number of particles are available, clustering centroids fail to match the quality of Stein-based particles. We take particles on the AG-News for an example. As illustrated in Fig. 3b, centroids are neither representative nor diverse. We attribute this to cluster collapse, caused by the lack of an explicit term to push the centroids apart. Fig. 3c provides a closer look at the locally grouped Stein-based particles but revealed that this area is dominated by cluster centroid which eventually confirms the consistent performance of Stein-based particles.

*RQ2: The necessity of coherence.* To verify whether coherence improves both understanding and reasoning tasks (We use ICL as a representative setting, as it is more sensitive to data quality.), we remove the refinement process and apply our method to four datasets. Fig. 4 shows the performance changes, from which we have the following observations. (i) Text understanding tasks also benefit from coherence, especially on the SST-2 dataset. (ii) Coherence is more critical for reasoning

Table 5: Evaluation with similarity metrics on four benchmarks. Abbreviations: Wass. (Wasserstein), MAU. (MAUVE score), Prec. (Precision), Rec. (Recall).

| Dataset | Methods | FID (↓) | KL (↓) | TV (↓) | Wass. (↓) | MAU. (↑) | Prec. (↑) | Rec. (↑) | F1 (↑) |
|---|---|---|---|---|---|---|---|---|---|
| Ag-News | Random | 0.7606 | **0.0411** | **0.0951** | 0.0187 | **0.9943** | **1.0000** | 0.9432 | 0.9708 |
| | DaLLME | 0.9454 | 0.1285 | 0.1922 | 0.0179 | 0.9541 | 0.4333 | 0.0357 | 0.0661 |
| | MGD³ | 0.8580 | 0.1203 | 0.1881 | 0.0256 | 0.9510 | 0.5333 | 0.0683 | 0.1211 |
| | Aug-PE | 0.9091 | 1.1767 | 0.3703 | 0.0487 | 0.6280 | 0.7500 | 0.0700 | 0.1281 |
| | PInR | **0.7135** | 0.0717 | 0.1426 | **0.0114** | 0.9836 | 0.7083 | 0.2521 | 0.3718 |
| SST-2 | Random | **0.6640** | **0.0169** | **0.0755** | **0.0097** | **0.9988** | **1.0000** | **0.8851** | **0.9391** |
| | DaLLME | 0.8744 | 1.6857 | 0.4875 | 0.1118 | 0.4502 | 0.2250 | 0.0935 | 0.1321 |
| | MGD³ | 0.7627 | 0.4694 | 0.3569 | 0.0735 | 0.6959 | 0.1500 | 0.1542 | 0.1521 |
| | Aug-PE | 0.8728 | 6.4459 | 0.6679 | 0.1497 | 0.1597 | 0.3625 | 0.0867 | 0.1400 |
| | PInR | 0.7665 | 0.6519 | 0.4438 | 0.0952 | 0.4691 | 0.2375 | 0.1339 | 0.1712 |
| GSM8K | Random | **0.0655** | 0.0530 | **0.1079** | 0.0016 | 0.9907 | **1.0000** | 0.8363 | 0.9108 |
| | DaLLME | 0.0889 | 0.0665 | 0.1324 | **0.0013** | 0.9857 | 0.9300 | 0.8170 | 0.8699 |
| | MGD³ | 0.0945 | 0.1939 | 0.1836 | 0.0019 | 0.9589 | 0.7900 | 0.7216 | 0.7542 |
| | Aug-PE | 0.2871 | 1.9482 | 0.5844 | 0.0158 | 0.2147 | 0.1700 | 0.7333 | 0.2760 |
| | PInR | 0.0948 | **0.0433** | 0.1145 | **0.0013** | **0.9933** | 0.9560 | **0.8793** | **0.9160** |
| Quora-QuAD | Random | 0.1736 | 0.1434 | 0.1319 | 0.0019 | 0.9829 | **1.0000** | 0.9141 | **0.9551** |
| | DaLLME | **0.1152** | **0.0141** | **0.0648** | 0.0014 | **0.9991** | 0.7980 | **0.9451** | 0.8653 |
| | MGD³ | 0.2045 | 0.1287 | 0.2084 | 0.0019 | 0.9498 | 0.3460 | 0.7955 | 0.4822 |
| | Aug-PE | 0.8884 | 9.1747 | 0.8404 | 0.0302 | 0.0249 | 0.1940 | 0.1046 | 0.1359 |
| | PInR | 0.1882 | 0.0448 | 0.1091 | **0.0011** | 0.9928 | 0.8480 | 0.8993 | 0.8729 |

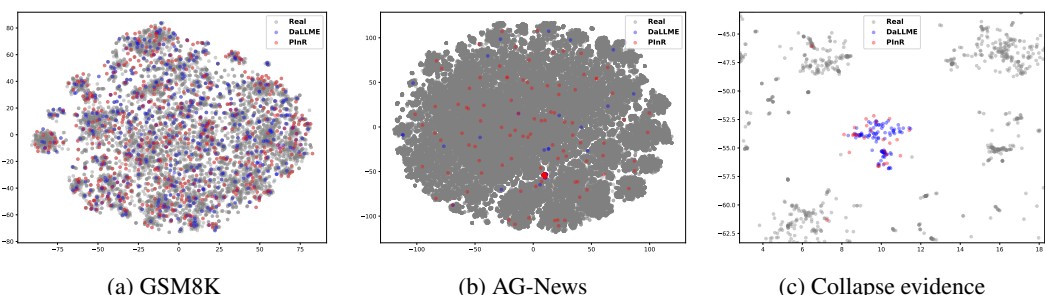

(a) GSM8K  (b) AG-News  (c) Collapse evidence

Figure 3: Particles visualization in the embedding space (zoomed in for better visualization).

tasks, the performance drop is significant across different model architectures. This agrees with our expectation as coherence directly affects sample usability in reasoning tasks.

*RQ3: Reliance on API.* Our method PInR and the baseline Aug-PE both employ third-party APIs to assist in generating condensed text. To evaluate the impact of this reliance, we compare them in terms of performance versus API cost. Fig. 5 presents the results, showing that across all tasks, PInR achieves better performance while incurring lower API costs. We attribute this advantage to the warm start provided by inverted text samples: rather than relying on the API to randomly guess plausible data samples, our method inverts informative particles from the embedding space, leveraging the model's generalization on the data manifold.

# 6 DISCUSSION

## 6.1 PRIVACY STUDY

One advantage of condensed data generation over coreset selection is that the original data remain private, and no raw samples need to be shared. However, this remains as a conceptual property and often lacks theoretical justification in practice. Therefore, we empirically assess potential leakage by first retrieving the most similar neighboring text and then computing bigram and unigram overlaps (Martin et al., 1998). Their scores are 0.4476 and 0.5935, respectively, with random selection yielding 1 for both as a reference. This indicates that condensed data shares partial tokens with the original data, which is expected since Stein-based particles tend to converge toward high-density

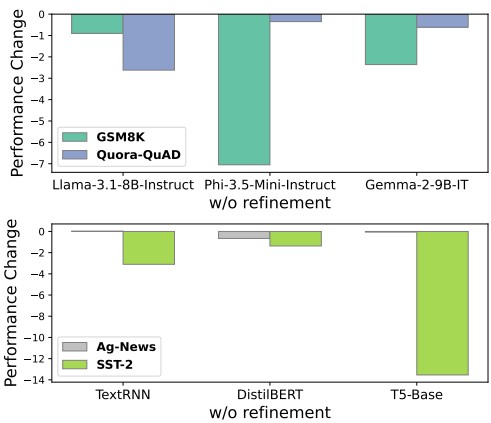
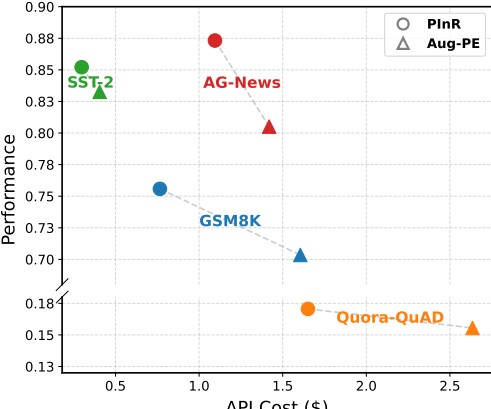

Figure 4: Performance change w/o refinement

Figure 5: Performance versus API cost.

regions and thus unavoidably lie close to real samples. A possible workaround is to manually constrain Stein-based particles to stay away from original samples, though this comes at the risk of sacrificing model performance.

When the original training set involves sensitive membership information, the condensation algorithm must satisfy differential privacy (DP) (Dong et al., 2022). This requirement serves as additional layer of privacy given that MaTC inherently mitigates risks of text content leakage. Our method of this version can be equipped with DP, while the direct apply may not be efficient, because we need to handle decoder training and score function (See Step 3 and 5 of the algorithm in Appendix A.2). Suppose the decoder is pre-trained. In that case, the lack of coherence in the inverted text may be compensated by invoking multiple rounds of API calls, reducing our method to Aug-PE in the extreme case.

### 6.2 LIMITATION

The sequence length of text data in our experiments cannot be very long. This design choice follows the observation of Morris et al. (2023) that training text decoders on long sequences is difficult. With less meaningful inverted long text, the proposed method may become unstable as refinement has to significantly revise text to align with particles rather than to guide generation toward the real data distribution. In addition, coherence is the key property we identify as essential for extending condensation to broader tasks. However, reframing highly complex structures, such as multi-turn dialogue (Li et al., 2017), remains difficult. For instance, when special tokens like [SEP] are not recovered, it requires advanced API to complete refinement.

## 7 CONCLUSION

Beyond understanding tasks, this work takes a step forward in generating condensed text samples tailored for reasoning-realated tasks. To the best of our knowledge, it is the first to explicitly identify three key properties that condensed text are expected to satisfy. Building on this insight, we proposed a two-stage method PInR that integrates informative Particle generation in embedding space with an Invert-and-Refinement (InR) procedure. By explicitly considering all three properties, our proposed method PInR generalizes effectively across both understanding and generation tasks. Extensive experiments on benchmark datasets demonstrate that our method consistently outperforms existing baselines, narrowing the gap between condensed and full-data training while retaining strong generalization to diverse downstream models. These findings highlight the importance of coherence-aware condensation and provide evidence that principled design of condensed samples can substantially benefit reasoning-oriented applications. We also discussed potential limitations, including the adaptability to long sequence or complex structured corpus, and outlined practical workarounds. We hope that this work lays the foundation for future research on condensation methods that are not only efficient but also faithful to the structural and semantic properties of natural language data.

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

## A  MORE DETAILS ABOUT PINR

### A.1  IMPLEMENTATION

For reasoning related datasets such as GSM8K and Quora-QuAD, we put a special token `[SEP]` as a separation of questions and answers. The inverted text is thus expected to recover the token so that it can be clearly treated as a natural textual sample for downstream evaluation. This is not the only choice, for example, one can add another token between contexts and questions. In this sense, if these tokens are not recovered, the APIs used in refinement is supposed to revise them.

Decoder training follows the vec2text model (Morris et al., 2023) which uses reconstruction loss with a recursive corrector. Note that this corrector mainly aims to align the sequences, which is different from our refinement process. At $r$-th iteration, it is written as

$$p(x_r|e) = \sum_{x_{r-1}} p(x_{r-1}|e)p(x_r|x_{r-1}, e), \tag{5}$$

where the second factor is parameterized as a conditional generator.

There are additional techniques that have been proposed to boost the quality of text condensation. For example, task-specific prompts can be applied to each sample before converting it into embeddings (Tao et al., 2024). We use the same prompt across different methods. Recent work has also considered sample difficulty (Azeemi et al., 2023) as a factor in condensation. However, this strategy is not strictly task-agnostic, and we leave this line of exploration to future work.

### A.2  ALGORITHM

---

**Algorithm 1** PInR

---
**Require:** Embedding model $\psi$, decoder $\omega$, original training set $\mathcal{T}_o$, seed number $K$, a callable API
  1: Initialize $\psi$ and $\omega$
  2: Obtain embeddings $e_i$ through $e_i = \psi(x_i)$ for each $x_i$ in $\mathcal{T}_o$
  3: Train score function $\nabla_e \log p(e)$ with all $e_i$
  4: Obtain particles $\{\tilde{e}\}^M$ with each updated by Eq. (2)
  5: Train decoder $\omega$ with $\{(x_i, e_i)\}^N$ pairs following (Morris et al., 2023)
  6: **for** $j = 1, \ldots, M$ **do**
  7:     Get $\tilde{x}_{j0} = \omega(\tilde{e}_j)$
  8:     $S_0 \leftarrow \{\tilde{x}_{j0}\}$
  9:     **for** $t = 0, \ldots, T$ **do**
 10:         Generate $L$ variants $\{a^0, a^1, ..., a^{L-1}\}$ for any $a \in S_t$ by calling API
 11:         **if** $t! = T$ **then**
 12:             $S_{t+1}$ is updated by the top-$K$ closest variants to $\tilde{e}_j$
 13:         **else**
 14:             Pick the most closest variant as $\tilde{x}_j$
 15:         **end if**
 16:     **end for**
 17: **end for**
 18: **return** Condensed samples $\{\tilde{x}_j\}^M$

---

### A.3  CONVERGENCE ANALYSIS

As shown in Fig. 2, our PInR framework is a two-stage method that trains a decoder and a score function while optimizing text in the neighborhood of fixed points. The Stein-based particles converge with theoretical guarantees as shown in (Liu & Wang, 2016), while the inversion model is applied based on its generalization ability, since Stein-based particles do not necessarily lie within the convex hull of the original samples. However, as mentioned earlier, if the inversion model is not sufficiently strong, we can resort to API-based refinement, which iteratively revises the text. By computing similarity with anchored particles and leveraging the theoretical results in (Lin et al., 2024), we show that our method overall converges.

Table 6: Training configurations on the AG-News dataset

| Downstream Model | DaLLME | MGD³ | Aug-PE | PInR |
|---|---|---|---|---|
| TextRNN | MAX_TOKENS = 6000000
BATCH_SIZE = 8
EMBEDDING_DIM = 100
DROPOUT = 0.5
NUM_EPOCHS = 20
LR = [5e-3]
HIDDEN_DIM=100
N_LAYERS=2
BIDIRECTIONAL=True
MAX_LEN = 128
USE_PRETRAINED = True | MAX_TOKENS = 6000000
BATCH_SIZE =2
EMBEDDING_DIM = 100
DROPOUT = 0.5
NUM_EPOCHS = 20
LR = [5e-3]
HIDDEN_DIM=100
N_LAYERS=2
BIDIRECTIONAL=True
MAX_LEN = 128
USE_PRETRAINED = True | MAX_TOKENS = 6000000
BATCH_SIZE = 8
EMBEDDING_DIM = 100
DROPOUT = 0.5
NUM_EPOCHS = 20
LR = [5e-3]
HIDDEN_DIM=100
N_LAYERS=2
BIDIRECTIONAL=True
MAX_LEN = 128
USE_PRETRAINED = True | MAX_TOKENS = 6000000
BATCH_SIZE = 32
EMBEDDING_DIM = 100
DROPOUT = 0.5
NUM_EPOCHS = 20
LR = [5e-3]
HIDDEN_DIM=100
N_LAYERS=2
BIDIRECTIONAL=True
MAX_LEN = 128
USE_PRETRAINED = True |
| DistilBERT | BATCH_SIZE = 8
MAX_LENGTH = 128
NUM_EPOCHS = 20
LR = 5e-5 | BATCH_SIZE = 8
NUM_EPOCHS = 20
MAX_LENGTH=128
LR = 5e-4 | BATCH_SIZE = 8
MAX_LENGTH = 128
NUM_EPOCHS = 20
LR = 5e-5 | BATCH_SIZE = 8
MAX_LENGTH = 128
NUM_EPOCHS = 20
LR = 5e-5 |
| T5-base | BATCH_SIZE = 8
NUM_EPOCHS = 20
MAX_LENGTH=128
LR = 5e-4 | BATCH_SIZE = 8
NUM_EPOCHS = 20
MAX_LENGTH=128
LR = 5e-5 | BATCH_SIZE = 8
NUM_EPOCHS = 20
MAX_LENGTH=128
LR =5e-4 | BATCH_SIZE = 8
NUM_EPOCHS = 20
MAX_LENGTH=128
LR = 5e-4 |

Table 7: Training configurations on the SST-2 dataset

| Downstream Model | DaLLME | MGD³ | Aug-PE | PInR |
|---|---|---|---|---|
| TextRNN | MAX_TOKENS = 6000000
BATCH_SIZE =2
EMBEDDING_DIM = 100
DROPOUT = 0.5
NUM_EPOCHS = 20
LR = [5e-4]
HIDDEN_DIM=100
N_LAYERS=2
BIDIRECTIONAL=True
MAX_LEN = 128
USE_PRETRAINED = True | MAX_TOKENS = 6000000
BATCH_SIZE =1
EMBEDDING_DIM = 100
DROPOUT = 0.5
NUM_EPOCHS = 20
LR = [2e-3]
HIDDEN_DIM=100
N_LAYERS=2
BIDIRECTIONAL=True
MAX_LEN = 128
USE_PRETRAINED = True | MAX_TOKENS = 6000000
BATCH_SIZE = 4
EMBEDDING_DIM = 100
DROPOUT = 0.5
NUM_EPOCHS = 20
LR = [1e-3]
HIDDEN_DIM=100
N_LAYERS=2
BIDIRECTIONAL=True
MAX_LEN = 128
USE_PRETRAINED = True | MAX_TOKENS = 6000000
BATCH_SIZE =2
EMBEDDING_DIM = 100
DROPOUT = 0.5
NUM_EPOCHS = 20
LR = [5e-4]
HIDDEN_DIM=100
N_LAYERS=2
BIDIRECTIONAL=True
MAX_LEN = 128
USE_PRETRAINED = True |
| DistilBERT | BATCH_SIZE = 1
NUM_EPOCHS = 20
MAX_LENGTH=128
LR = 1e-5 | BATCH_SIZE = 8
NUM_EPOCHS = 20
MAX_LENGTH=128
LR = 5e-5 | BATCH_SIZE = 8
NUM_EPOCHS = 20
MAX_LENGTH=128
LR = 2e-5 | BATCH_SIZE = 1
NUM_EPOCHS = 20
MAX_LENGTH=128
LR = 1e-5 |
| T5-base | BATCH_SIZE = 1
NUM_EPOCHS = 20
MAX_LENGTH=128
LR = 5e-4 | BATCH_SIZE = 2
NUM_EPOCHS = 20
MAX_LENGTH=128
LR = 5e-4 | BATCH_SIZE = 8
NUM_EPOCHS = 20
MAX_LENGTH=128
LR = 5e-4 | BATCH_SIZE = 1
NUM_EPOCHS = 20
MAX_LENGTH=128
LR = 1e-5 |

# B  MORE DETAILS ABOUT EXPERIMENTS

## B.1  EXPERIMENTAL SETUP

The choice of embedding model $\psi(\cdot)$. Recent works which involve embedding space typically use sentence transformer or language model embeddings. As "text-embedding-ada-002" has been identified powerful in (Tao et al., 2024; Xie et al., 2024), we use it throughout our experiments without further exhaustively testing other alternatives. For all datasets, we generate 3 variants, i.e. $L = 3$ and set the seed number $K$ as 1.

Here we give the prompts we used for refinement.

**GSM8K**:

*You are a math tutor. Your job is to correct flawed reasoning in following math Q&A. Always output the corrected Q&A in the following exact format. Do not add explanations or extra text. Format: Q: *corrected question text* A: *corrected answer*. Input Q: {question} Input A: {answer}*

**Quora-QuAD**:

*You are a helpful assistant that writes Q&A pairs in the style of Quora. Your job is to make sentences fluent, grammatically correct, and logically coherent. Write questions and answers in the natural, conversational, and explanatory style of Quora, where questions are natural, curious and clear, and answers are clear, concise, conversational, thoughtful, detailed, and easy to understand. Return only the corrected Q&A, nothing else. Format: Q: \*corrected question text\* A: \*corrected answer\* Input Q: {question} Input A: {answer}*

**Ag-News**:

*You will be given a piece of News text, The text may be grammatically incorrect, awkward, incomplete, or unnatural. Your task is to polish and rewrite the text. Please polish the following text into fluent, coherent English that reads like a professional {category} news report, completing unclear expressions while preserving the original meaning. Input Text: {text}*

**SST-2**:

*You will be given a piece of sentence in movie review, The sentence may be incorrect, awkward, incomplete, or unnatural. Your task is to polish and rewrite the it. Please polish the following sentence into fluent, coherent English that reads like convey a {category} sentiment, rewrite the unclear expressions while preserving the original words and meaning as much as possible. Input Sentence: {sentence}*

Additionally, following Tao et al. (2024), we applied task-specific prompts to the classification tasks before converting the data into embeddings.

**Ag-News**:

*Read the following news article and classify it into one of our categories: World, Sports, Business, or Science/Technology. Provide a brief rationale for your classification.*

**SST-2**:

*Read the following sentences and classify it as either positive or negative sentiment. Provide a brief rationale for your classification.*

We also provide downstream model configurations in Tables 6 and 7 for reproducing the reported results.

### B.2 EXPERIMENTAL RESULTS

For Quora-QuAD dataset, we report the experimental results in terms of the other three metrics in Table 8, 9 and 10.

We also test the possibility of using a pretrained model for the decoder. Here we preset an example for GSM8K using a pretrained model installed from `https://github.com/vec2text/vec2text`.

*"Donny is a book reader and she has a book for the whole week. Donny is a book reader and she has a book for the whole week. Donny is a book reader and she has a book for the whole week. Donny is a book reader and she has a book for the whole week. The number of books he can get is: 2/5 = 5/5 = 2/5 = 2/5 = 2/5 = 2/5 = 2', '(10) If a rabbit has come out of the cage in 20 minutes, and the rabbits have come out of the cage in 30 minutes, the rabbits will have come out of the cage in 20 minutes."*

One can see there are many repetitive sentences as well as non-logical reasoning paths. For advanced APIs like ChatGPT-5, it is not hard to revise into coherent Q&A samples, while this will become expensive if we do multi-step refinement to align with the optimized particles.

Table 8: Evaluation on Quora-QuAD Dataset in terms of Rouge2

| Downstream Model | Zero-shot | Type | Random | DaLLME | MGD$^3$ | Aug-PE | PInR |
|---|---|---|---|---|---|---|---|
| Llama-3.1-8B-Instruct | $2.84_{\pm0.00}$ | FT | $2.92_{\pm0.00}$ | $\underline{2.94}_{\pm0.00}$ | $2.85_{\pm0.00}$ | $2.92_{\pm0.00}$ | $\mathbf{2.98}_{\pm\mathbf{0.02}}$ |
| | | ICL | $\underline{2.87}_{\pm0.10}$ | $2.46_{\pm0.05}$ | $2.52_{\pm0.01}$ | $2.82_{\pm0.03}$ | $\mathbf{3.02}_{\pm\mathbf{0.03}}$ |
| Phi-3.5-Mini-Instruct | $2.12_{\pm0.00}$ | FT | $\mathbf{2.15}_{\pm\mathbf{0.00}}$ | $2.07_{\pm0.00}$ | $2.10_{\pm0.00}$ | $\mathbf{2.15}_{\pm\mathbf{0.00}}$ | $\mathbf{2.15}_{\pm\mathbf{0.04}}$ |
| | | ICL | $2.26_{\pm0.10}$ | $\mathbf{2.55}_{\pm\mathbf{0.04}}$ | $2.51_{\pm0.04}$ | $2.41_{\pm0.01}$ | $\mathbf{2.55}_{\pm\mathbf{0.06}}$ |
| Gemma-2-9B-IT | $0.94_{\pm0.01}$ | FT | $\mathbf{0.96}_{\pm\mathbf{0.00}}$ | $0.89_{\pm0.00}$ | $0.92_{\pm0.00}$ | $0.90_{\pm0.00}$ | $\underline{0.94}_{\pm0.01}$ |
| | | ICL | $1.77_{\pm0.06}$ | $1.82_{\pm0.01}$ | $1.83_{\pm0.04}$ | $\mathbf{1.88}_{\pm\mathbf{0.00}}$ | $\underline{1.87}_{\pm0.08}$ |

Table 9: Evaluation on Quora-QuAD Dataset in terms of RougeL

| Downstream Model | Zero-shot | Type | Random | DaLLME | MGD$^3$ | Aug-PE | PInR |
|---|---|---|---|---|---|---|---|
| Llama-3.1-8B-Instruct | $10.35_{\pm0.00}$ | FT | $10.31_{\pm0.00}$ | $\mathbf{10.46}_{\pm\mathbf{0.00}}$ | $10.24_{\pm0.01}$ | $10.28_{\pm0.01}$ | $\underline{10.44}_{\pm0.00}$ |
| | | ICL | $10.09_{\pm0.15}$ | $9.30_{\pm0.04}$ | $09.20_{\pm0.14}$ | $\underline{10.52}_{\pm0.13}$ | $\mathbf{10.86}_{\pm\mathbf{0.05}}$ |
| Phi-3.5-Mini-Instruct | $7.92_{\pm0.01}$ | FT | $8.06_{\pm0.01}$ | $8.04_{\pm0.01}$ | $\underline{8.08}_{\pm0.01}$ | $8.03_{\pm0.01}$ | $\mathbf{8.12}_{\pm\mathbf{0.03}}$ |
| | | ICL | $8.68_{\pm0.11}$ | $\mathbf{9.66}_{\pm\mathbf{0.05}}$ | $9.58_{\pm0.04}$ | $\underline{9.62}_{\pm0.02}$ | $9.53_{\pm0.04}$ |
| Gemma-2-9B-IT | $4.26_{\pm0.01}$ | FT | $\mathbf{4.30}_{\pm\mathbf{0.01}}$ | $4.22_{\pm0.00}$ | $4.23_{\pm0.01}$ | $4.18_{\pm0.01}$ | $\underline{4.27}_{\pm0.01}$ |
| | | ICL | $8.12_{\pm0.09}$ | $\underline{8.26}_{\pm0.04}$ | $\mathbf{8.30}_{\pm\mathbf{0.01}}$ | $8.24_{\pm0.02}$ | $8.21_{\pm0.01}$ |

Table 10: Evaluation on Quora-QuAD Dataset in terms of RougeLsum

| Downstream Model | Zero-shot | Type | Random | DaLLME | MGD$^3$ | Aug-PE | PInR |
|---|---|---|---|---|---|---|---|
| Llama-3.1-8B-Instruct | $11.75_{\pm0.01}$ | FT | $11.74_{\pm0.00}$ | $\mathbf{12.00}_{\pm\mathbf{0.00}}$ | $11.62_{\pm0.01}$ | $11.71_{\pm0.01}$ | $\underline{11.93}_{\pm0.03}$ |
| | | ICL | $\underline{11.86}_{\pm0.23}$ | $10.89_{\pm0.04}$ | $10.72_{\pm0.25}$ | $11.81_{\pm0.18}$ | $\mathbf{12.69}_{\pm\mathbf{0.07}}$ |
| Phi-3.5-Mini-Instruct | $8.95_{\pm0.01}$ | FT | $9.05_{\pm0.01}$ | $9.05_{\pm0.01}$ | $\underline{9.10}_{\pm0.01}$ | $9.05_{\pm0.01}$ | $\mathbf{9.14}_{\pm\mathbf{0.03}}$ |
| | | ICL | $9.66_{\pm0.15}$ | $\mathbf{10.94}_{\pm\mathbf{0.06}}$ | $\underline{10.82}_{\pm0.00}$ | $10.74_{\pm0.00}$ | $10.79_{\pm0.05}$ |
| Gemma-2-9B-IT | $4.56_{\pm0.00}$ | FT | $\mathbf{4.61}_{\pm\mathbf{0.00}}$ | $4.54_{\pm0.01}$ | $4.55_{\pm0.00}$ | $4.49_{\pm0.00}$ | $\underline{4.58}_{\pm0.01}$ |
| | | ICL | $8.81_{\pm0.08}$ | $\underline{8.98}_{\pm0.08}$ | $\mathbf{9.04}_{\pm\mathbf{0.00}}$ | $8.94_{\pm0.03}$ | $8.95_{\pm0.01}$ |

