# OpenReview forum: "Model-Agnostic Text Condensation with Coherence Awareness"
_ICLR.cc/2026/Conference — ICLR 2026 Conference Withdrawn Submission_

### Official Review · Reviewer_brc7 · 2025-10-28

**Soundness:** 3
**Presentation:** 3
**Contribution:** 2
**Rating:** 4
**Confidence:** 4

**Summary:**

The authors propose a novel method for text condensation, where the original dataset is converted into embeddings, condensed samples are generated in the embedding space, and these condensed samples are inverted back into text. The proposed method contains two main contributions. First, it introduces a novel condensed sample generation technique that optimizes condensed samples to make their distribution closer to that of the original samples. Second, they propose an invert-and-refine approach using an API to improve the coherence of the inverted text. Experiments demonstrate that the proposed method achieves the best performance compared to existing text condensation methods and core set selection methods.

**Strengths:**

1. The authors focus on coherence in text condensation, which is a crucial aspect of inference tasks. While DaLLME (Tao et al., 2024) considered the readability of condensed samples, it did not address coherence. This paper is the first to explicitly take coherence into account.

2. The authors propose a method to optimize condensed samples so that their distribution becomes closer to that of the original samples, introducing an invert-and-refine approach using an API. These methods are simple yet intuitive and contribute to performance improvements.

3. The proposed method achieves superior performance compared to existing approaches, including coreset selection (Random) and three condensation methods (DaLLME, MGD3, and Aug-PE). Text condensation is a particularly promising technique in an era of rising model training costs, and its potential impact is substantial.

**Weaknesses:**

1. The authors should conduct a more comprehensive comparison with coreset selection methods. Although their method addresses the issue of coherence loss, this problem arises specifically from generating synthetic data. In contrast, coreset selection methods do not suffer from this issue because the samples are drawn from real training data, which inherently ensures coherence. In particular, (Maekawa et al., 2025b) demonstrated that K-centers and Herding outperform random selection as coreset selection strategies. To clarify the importance of generating synthetic data, the authors should include comparisons with K-centers and Herding.

2. Similarly, the authors should compare INVERT-AND-REFINE with a simple nearest-neighbor search. Specifically, they should evaluate performance by selecting the real data point closest to the optimized embedding, which can be formulated as: $\tilde{x_j} = \arg\min_{x} d(\psi(x),  \tilde{e_j}) \quad \text{s.t. } x \in \mathcal{T}_o$. Since this approach selects samples directly from the real training set, it naturally ensures coherence. Evaluating this strategy would help clarify the importance of generating synthetic data.

3. The difference between the condensed samples optimization method represented by Equation (2) and existing condensed samples optimization methods is not explained. The authors should describe what condensed samples optimization methods DaLLME, MGD3, and Aug-PE employ and explain why the condensed samples optimization method represented by Equation (2) is superior to these existing methods.

4. The compared method MGD3 is a condensation method for images. The authors should clarify how they are applying this to text condensation.

**Questions:**

Please add explanations regarding the weaknesses.

---

### Official Review · Reviewer_P9Pv · 2025-10-29

**Soundness:** 1
**Presentation:** 3
**Contribution:** 2
**Rating:** 2
**Confidence:** 4

**Summary:**

This paper addresses the problem of model-agnostic text condensation (MaTC) by introducing "coherence" as a critical property, arguing it is more stringent than mere "readability" and essential for downstream tasks. The paper then proposes PInR, a two-stage framework to generate a small dataset that embodies three key properties: representativeness, diversity, and coherence. The first stage optimizes a set of "particles" in the semantic embedding space using a Stein-based method to ensure representativeness and diversity. The second stage, Invert-and-Refine (InR), decodes these particles into initial text and then uses an API-assisted refinement process to enforce coherence while ensuring the final text remains semantically faithful to the optimized particle. Experiments across text understanding and reasoning benchmarks demonstrate that PInR outperforms existing SOTA baselines.

**Strengths:**

1. The PInR framework is logically designed. It decouples the problem by first optimizing for representativeness, diversity in a continuous embedding space and then separately enforcing coherence in the discrete text space.
2. The ablation study provides empirical support for the paper's central hypothesis. Removing the refinement step leads to a drastic performance degradation, especially on reasoning datasets, confirming the necessity of this component.

**Weaknesses:**

1. The framework's effectiveness is critically dependent on powerful, black-box APIs at every stage. It uses text-embedding-ada-002 as the encoder and outsources its core "coherence" contribution to a strong generative API. Without any ablations on open-source models, it's impossible to tell if the PInR method is good, or if the APIs are just doing all the work.
2. The "warm start" efficiency argument seems to be flawed. The paper's own inverted text example in Appendix B.2 is nonsensical and logically flawed. This suggests the API isn't refining this input but is just ignoring it and re-generating from scratch, which invalidates the entire efficiency claim.
3. The Random baseline is exceptionally strong and, in several ICL settings for reasoning, it outperforms PInR. Given that Random is computationally trivial, the practical value of this complex, multi-stage, API-dependent synthesis method is questionable.

**Questions:**

Authors should give their response and explanation with respect to  the weaknesses mentioned above.

---

### Official Review · Reviewer_jok2 · 2025-10-30

**Soundness:** 2
**Presentation:** 2
**Contribution:** 3
**Rating:** 4
**Confidence:** 3

**Summary:**

This paper proposes PInR, a model-agnostic framework for text dataset condensation. It optimizes representative and diverse particles in the embedding space using Stein variational principles and then converts them into coherent text through an invert-and-refine process. The key novelty lies in explicitly enforcing textual coherence, beyond readability, to improve reasoning-oriented tasks. Experiments on AG-News, SST-2, GSM8K, and Quora-QuAD show that PInR consistently outperforms existing methods, achieving better accuracy and distributional similarity with strong robustness across both understanding and reasoning benchmarks.

**Strengths:**

1. The paper is well-motivated and introduces coherence as a fundamental property in text condensation.
2. Its theoretical foundation using Stein variational optimization is principled and general.
3. The experimental evaluation is extensive and convincing, showing clear improvements across tasks.
4. The framework is modular and reusable and bridges embedding-based optimization with text-level refinement effectively.

**Weaknesses:**

1. The method relies heavily on external LLM APIs for refinement, which challenges reproducibility and model-agnostic claims.
2. The definition of coherence is heuristic rather than formally measurable.
3. Evaluation is limited to medium-scale datasets, and the approach may struggle with longer or multi-turn sequences.
4. Privacy and efficiency analyses are discussed but not rigorously demonstrated.

**Questions:**

1. How sensitive is the refinement quality to the specific API used, and could weaker open-source models achieve similar results?
2. How might the framework handle long or multi-turn textual data where coherence is more complex?

---

### Official Review · Reviewer_tizo · 2025-11-02

**Soundness:** 2
**Presentation:** 3
**Contribution:** 2
**Rating:** 6
**Confidence:** 3

**Summary:**

The paper introduces PInR, a novel framework for model-agnostic text condensation that explicitly integrates coherence as a key property alongside representativeness and diversity. The method optimizes “informative particles” in a semantic embedding space and converts them back into discrete text through an Invert-and-Refine (InR) process using an API-based coherence refinement. Experiments on understanding (AG-News, SST-2) and reasoning (GSM8K, Quora-QuAD) tasks show consistent gains over prior methods like DaLLME, MGD3, and Aug-PE. The authors also analyze privacy implications, computational cost, and ablation on coherence.

**Strengths:**

1. Introducing coherence as an explicit property in text condensation is both intuitive and impactful, especially for reasoning tasks. It moves beyond the typical focus on readability and directly links data quality to logical soundness.
2. The two-stage PInR pipeline is well-motivated and mathematically grounded. The use of kernelized updates and variational inference offers a principled way to approximate text distributions.
3. PInR does not depend on specific downstream architectures, and experiments span both classification and reasoning models.

**Weaknesses:**

1. The refinement stage relies on commercial LLM APIs (e.g., GPT-3.5), which raises questions about reproducibility and fairness of comparison, even though the authors mention cost control.
2. The Stein particle optimization in high-dimensional embedding spaces may become computationally expensive. Runtime or memory analyses are missing, and comparisons with lighter methods (e.g., clustering-based condensation) on efficiency would be valuable.

**Questions:**

See above.

---

### Note · Authors · 2025-11-21

I have read and agree with the venue's withdrawal policy on behalf of myself and my co-authors.